# Tetrodotoxin Profiles in Xanthid Crab *Atergatis floridus* and Blue-Lined Octopus *Hapalochlaena* cf. *fasciata* from the Same Site in Nagasaki, Japan

**DOI:** 10.3390/toxins15030193

**Published:** 2023-03-03

**Authors:** Yuchengmin Zhang, Yuta Yamate, Takeshi Takegaki, Osamu Arakawa, Tomohiro Takatani

**Affiliations:** Graduate School of Fisheries and Environmental Sciences, Nagasaki University, 1-14, Bunkyo-Machi, Nagasaki 852-8521, Japan

**Keywords:** tetrodotoxin, tetrodotoxin analogs, xanthid crab, blue-lined octopus

## Abstract

The xanhid crab *Atergatis floridus* and the blue-lined octopus *Hapalochlaena* cf. *fasciata* have long been known as TTX-bearing organisms. It has been speculated that the TTX possessed by both organisms is exogenously toxic through the food chain, since they are reported to have geographic and individual differences. The source and supply chain of TTX for both of these organisms, however, remain unclear. On the other hand, since crabs are one of the preferred prey of octopuses, we focused our attention on the relationship between the two species living in the same site. The aim of this study was to determine TTX concentrations and TTX profiles of *A. floridus* and *H.* cf. *fasciata*, collected simultaneously in the same site, and examine the relationship between them. Although there were individual differences in the TTX concentration in both *A. floridus* and *H.* cf. *fasciata,* the toxin components commonly contained 11-*nor*TTX-6(*S*)-ol in addition to TTX as the major components, with 4-*epi*TTX, 11-deoxyTTX, and 4,9-anhydroTTX as the minor components. The results suggest that octopuses and crabs in this site acquire TTX from common prey, including TTX-producing bacteria and/or may have a predator–prey relationship.

## 1. Introduction

Tetrodotoxin (TTX) is a potent neurotoxin first identified in pufferfish [1,2] and then subsequently identified in various other organisms, such as octopuses, newts, frogs, crabs, ribbon worms, flatworms, annelids, and bacteria [3,4,5,6,7,8,9,10]. TTX is a low molecular-weight neurotoxin that blocks voltage-gated sodium channels [11]. More than 20 analogs of TTX were found in marine and terrestrial animals [12,13]. Some of these analogs are highly toxic, especially 11-oxoTTX, which has a higher specific toxicity than TTX [14].

The blue-lined octopus *Hapalochlaena* cf. *fasciata* is a small octopus that has been reported to be distributed in Australia, Korea, and Japan [15,16,17,18]. The blue-ringed octopus contains TTX throughout the body, including the posterior salivary glands (PSGs), gonads, and arms, with the PSGs being the most highly toxic organ [15,19,20]. It has been considered that the octopus of the genus *Hapalochlaena* possesses TTX for feeding and defense [15]. Recently, Yamate et al. analyzed *H.* cf. *fasciata* collected in different areas of Japan and found individual and seasonal differences in the TTX concentrations [17].

In Japan, three species of toxic xanthid crabs (*Zosimus aeneus*, *Atergatis floridus*, and *Platypodia granulosa*) are reported to be highly toxic [6,21]. *A. floridus* is usually found on live coral reefs and rocks. The crab contains TTX and/or paralytic shellfish toxins (PSTs) depending on the habitat, and the appendages seem to be the most toxic part of the crab [22,23]. The toxins in toxic xanthid crabs are considered to be used for defense [23]. Many investigations of the toxin concentrations and toxin profiles of these crabs suggest a geographic distribution, even in a single reef [24,25,26].

Like pufferfish—although not all pufferfish species can accumulate TTX in large amounts—the most well-known marine organisms containing TTX are thought to acquire the neurotoxins directly and/or indirectly through food webs originating from TTX-producing bacteria [9,11]. Because of the individual and geographic differences mentioned above, octopuses and crabs are also assumed to accumulate TTX from prey organisms and/or TTX-producing bacteria which were isolated in the PSGs of octopuses and the intestines of *A. floridus* [9]. The TTX-origin organisms of blue-lined octopus, however, have not been reported, and the only known origin organism of the xanthid crab toxins on the reefs of Ishigaki Island is *Jania* sp. [10,27]. It is generally considered that the common octopus (*Octopus* sp.) preys on small crabs, and some laboratory experiments revealed that crabs are the highly preferred prey [28,29,30]. The relevance of the food chain between the blue-lined octopus and toxic crabs, however, has not been identified.

In the present study, we collected the xanthid crab *A. floridus* and blue-lined octopus *Hapalochlaena* cf. *fasciata* simultaneously from the same site. TTX concentration and TTX analogs in crabs and octopuses were analyzed to determine the relationship between these two TTX-bearing organisms and speculate the route of TTX toxification.

## 2. Results

### 2.1. Toxin Concentrations of A. floridus in 2020

The whole-body TTX concentration in *A. floridus* collected in December 2020 is shown in Table 1. TTX was detected in all nine specimens (mean weight: 7.53 g). The maximum TTX concentration in the nine crab samples was 45.46 μg/g, and the highest total TTX was 421.88 μg. The minimum TTX concentration was 0.60 μg/g (No. 6), and the lowest total TTX amount was 2.16 μg (No. 3). No significant correlation was found between body weight and the TTX concentration or body weight and the total TTX amount (r = 0.48 and 0.59; *p* = 0.19 and 0.09, respectively; n = 9 for both. R Studiov.4.1.0 (R Core Team 2021)). In the toxic fraction after activated charcoal treatment, the toxicity calculated from the amount of TTX by instrumental analysis was 1000 MU, whereas the actual score obtained by mouse assay was 2160 MU, suggesting the presence of other toxic components with TTX in the extract. Six saxitoxins (STXs) and six gonyautoxins (GTXs) were also analyzed for PSTs commonly detected in toxic crabs. However, all 12 PST components were found to be below the detection limit (<0.02 μg/g) (Appendix A). Figure 1 shows a chromatogram of the toxic fraction after activated charcoal treatment analyzed by high-performance liquid chromatography with fluorescence detection (HPLC-FLD), revealing six peaks that appeared to be TTX and TTX analogs. Peaks with 12.8-min, 15.3-min, and 16.4-min retention times on the HPLC chromatogram corresponded well to those of TTX (a), 4-*epi*TTX (b), and 4,9-anhydro TTX (c); unidentified peaks uk-1, uk-2, and uk-3 were detected at 14.5 min, 17.3 min, and 18.9 min. Purified and isolated TTX and uk-1 showed distinct peaks at *m*/*z* 320 > 162 (Rt: 6.95 min) and *m*/*z* 290 > 272 (Rt: 7.74 min), respectively, in liquid chromatography-tandem mass spectroscopy (LC-MS/MS) analysis (Figure 2). Based on the retention time of the peak in the MS chromatogram and the pattern of the two major ion spectra (*m*/*z* 290.09 [M + H]^+^ and *m*/*z* 272.09 [M + H-H_2_O]^+^), uk-1 was determined to be 11-norTTX-6(S)-ol (Figure 2). On the other hand, uk-2 and uk-3 could not be identified because no peak matching the MS spectrum of known TTX analogs was obtained in the LC-MS/MS analysis.

### 2.2. Toxin Concentrations and Profiles of A. floridus and Blue-Lined Octopus H. cf. fasciata in 2021

TTX concentrations in the appendages of *A. floridus* collected in 2021 are shown in Table 2. Low TTX concentrations were detected in the appendages of *A. floridus*. In the 11 *A. floridus* specimens collected in September 2021, the lowest concentration was 0.09 μg/g and the highest concentration was 9.77 μg/g (mean: 1.37 ± 2.8 μg/g), and in the four specimens collected in November, one specimen had a high concentration of 25.54 μg/g, while the other three specimens had low concentrations of 0.09 μg/g, 0.15 μg/g, and 0.39 μg/g (mean: 6.54 ± 12.7 μg/g).

All five specimens of *H.* cf. *fasciata* contained TTX in the PSGs and/or some other tissues. In the sample collected in September (No. 1), 1.62 μg/g of TTX was detected in the PSGs only and not in any other tissues (Table 3). In the four specimens collected in November, PSGs had the highest concentrations of TTX, ranging from 4.49 to 100.65 μg/g, and the other tissues contained almost the same lower concentrations of TTX. In specimen Nos. 2, 3, and 4, TTX was detected in all tissues, and the total amount of TTX was high (31.44 μg, 16.96 μg, and 19.72 μg, respectively), whereas in specimen No. 5, TTX was detected only in the PSGs and buccal mass, and the total amount of TTX was low (0.62 μg). 

In addition, specimen Nos. 2, 3, and 4 contained the highest total TTX amount in the arms among the tissues. LC-MS/MS and HPLC-FLD analysis were also used to examine TTX analogs in an extract from *A. floridus* and octopus *H*. cf. *fasciata*. Figure 3 shows HPLC-FLD chromatograms of crab extract collected in 2021 and of each of the six octopus tissues. In common with extracts of *A. floridus* and six tissues of *H*. cf. *fasciata*, the peaks of (a) TTX and (b) 11-*nor*TTX-6 (*S*)-ol were identified at retention times of 12.8 min and 14.5 min, respectively. The relative concentration of 11-*nor*TTX-6(*S*)-ol was 18 mol%, 43 mol%, 30 mol%, 11 mol%, and 8 mol% in the PSGs of five octopuses and 34 mol% in *A. floridus* extracts, which was calculated by assuming that the intensity of the fluorescence response is the same in all TTX. In addition, 4-*epi*TTX, and 4,9-anhydroTTX were also detected in the PSGs and buccal mass in *H.* cf. *fasciata* and extracts of *A. floridus*. LC-MS/MS analysis in MRM mode identified TTX and 4-*epi*TTX (*m*/*z* 320 > 162), 4,9-anhydroTTX (*m*/*z* 302 > 162), and 11-*nor*TTX-6(*S*)-ol (*m*/*z* 290 > 272), respectively (Figure 4). In addition, a peak of 11-deoxyTTX (not confirmed by HPLC-FLD analysis) was detected at *m*/*z* 304 > 162. On the other hand, two unknown peaks (uk-2 and uk-3) detected by HPLC-FLD analysis were not matched to known TTX analogs by LC-MS/MS analysis.

## 3. Discussion

In this study, we investigated the TTX concentration and profiles of the xanthid crab *A. floridus* and the blue-lined octopus *H.* cf. *fasciata*, which were collected simultaneously from the same site. A low level of TTX (0.09–25.54 μg/g) was detected in the appendages of crabs collected in 2021 with individual differences. According to their habitats and individual characteristics, the regional distributions of the toxin components, concentrations, and amounts differ among toxic xanthid crabs around the world. Highly toxic specimens are found on Ryukyu Island in Japan and Australia, and their toxin profiles consist almost exclusively of PSTs [25,31]. In contrast, on the mainland of Japan, in Taiwan, and on the islands of Cebu in the Philippines, xanthid crabs are reported to have low toxicity, with TTX being the main component [23,32,33]. In the Nagasaki waters, where this study was conducted, the toxin composition of *A. floridus* exhibits a TTX-dominant pattern similar to that of many specimens in mainland Japan, and no PSTs components were detected at all. These reports of regional differences in the toxin components and toxicity levels in xanthid crabs suggest that the TTX comes from the food chain including TTX-producing bacteria.

On the other hand, large individual differences in the toxicity of *H.* cf. *fasciata* are reported in various regions of Japan in terms of the TTX concentration [17]. We also found large individual differences in the amount of TTX possessed by *H.* cf. *fasciata* as well as differences in the TTX concentrations and total TTX amounts among five individual *H.* cf. *fasciata* collected from the same location (four of which were collected on the same day). In the *H.* cf. *fasciata* specimens examined in this study, TTX concentrations were higher mainly in the PSGs and buccal mass, while total TTX amounts were higher mainly in the arms, lateral mantle, and PSGs. These results are consistent with previous reports from Kyushu specimens and Australian specimens. The TTX levels in present study specimens were almost identical to those of the Kyushu (Fukuoka and Kumamoto) populations reported by Yamate et al. but with lower levels of TTX compared with the Australian specimens [15,17]. In the present specimens, the higher octopus weight resulted in higher TTX concentrations in the PSG, but an insufficient number of samples and a lack of correlation between PSG weight and body weight did not allow us to determine a relationship between the two. However, it may also simply be due to differences in the amount of TTX-bearing organisms consumed. Regardless of the individual differences in the TTX concentration, consistent with previous reports [15,17], the PSGs of all the specimens had the highest TTX concentration compared with the other tissues, and the amount of TTX was higher in the arms. Based on previous reports and results of the present study, it was inferred that the TTX in the PSGs of *H.* cf. *fasciata* is used for attack, and the TTX in the arms is used for defense [15,17,34].

In this study, instrumental analysis of the toxin components of *A. floridus* revealed that TTX and 11-*nor*TTX-6(*S*)-ol were the major components, with 4-*epi*TTX, 11-deoxyTTX, and 4,9-anhydroTTX as minor components. From previous reports, TTX-related components such as 11-oxoTTX, 11-*nor*TTX-6(*R*)-ol, and 11-saxitoxinethanoic acid are found in *A. floridus* in addition to TTX, 4-*epi*TTX, and 4,9-anhydroTTX [35,36]. On the other hand, there are few reports of TTX analogs in toxic octopuses. Asakawa et al. detected 4-*epi*TTX, 6-*epi*TTX, and 4,9-anhydroTTX in the greater blue-ringed octopus *Hapalochlaena lunulata* [20]. In the present study, blue-lined octopus *H.* cf. *fasciata* also contained non-equilibrium TTX analogs such as 11-*nor*TTX-6(*S*)-ol and 11-deoxyTTX that were somewhat similar to the toxin components of crabs collected at the same time (Figure 3 and Figure 4).

Unknown peaks such as uk-2 and uk-3 were detected by HPLC-FLD analysis, and they may be toxic components because the toxicity calculated by LC-MS/MS analysis (converted from the amount of TTX) and the toxicity determined by the mouse bioassay differed significantly. On the other hand, because some analogs such as deoxyTTXs have very low fluorescence intensities, LC/MS or LC-MS/MS analysis was also required [37]. In order to clarify the details of the toxification mechanism of both organisms, quantitative analysis of not only 11-*nor*TTX-6(*S*)-ol and 11-deoxyTTX found in this study but also other non-equilibrium analogs such as trideoxy and dideoxy analogs is required after the purification of toxic components and the preparation of quantification standards.

This is the first study on the TTX concentration and profiles of the xanthid crab *A. floridus* and the blue-lined octopus *H.* cf. *fasciata* collected from the same site. The similar toxin profiles of the two species suggest that the similar TTX-bearing prey organisms living in this site are shared by *A. floridus* and *H.* cf. *fasciata* as the source of TTX. This suggestion is supported by reports that the possession of similar major toxins in several different organisms, such as two species of ribbon worm [38], pufferfish and goby [39], suggests the presence of a common prey. TTX is thought to accumulate in higher trophic levels through the food chain starting from the TTX-producing bacteria [7,9]. Since TTX-producing bacteria were found in the bodies of most TTX-bearing species including the *A. floridus* and blue-ringed octopus [9], perhaps microbial organisms may be one of the sources of the toxin. In addition, TTX-producing bacteria have also been found in marine sediments [40,41], and it is possible that free-living bacteria at the same site may cause similar TTX profiles. In recent work, it was found that the spore culture of the marine bacterium Bacillus sp. 1839 contains a non-equilibrium analog of the 5,6,11-trideoxyTTX [42]. To date, however, non-equilibrium analogs other than TTX and 5,6,11-trideoxyTTX have not been reported in bacteria [9], and the productivity of TTX-related analogs needs to be confirmed in future studies. It is also necessary to experimentally prove the presence or absence of chemical conversion of TTX and its non-equilibrium analogs in the bodies of crabs and octopuses.

Octopuses are active predators, preying on a variety of benthic animals, including crabs, bivalves, gastropods, and fish [43]. In contrast, crabs are omnivorous, and it was reported that red algae (*Hypnea* sp.), bivalves, gastropods, animal tissues, and sand were preferentially found in the stomach contents of *A. floridus* collected on the Miura Peninsula in Japan [44], so it is not surprising that octopuses and *A. floridus* feed on the same food. There are many reports about organisms of lower trophic levels, such as snails, ribbon worms, and flatworms, around the world that contain TTX and non-equilibrium analogs [38,45,46,47,48]. Since bacteria and lower trophic levels of organisms were not investigated in this study, it is necessary to search for organisms that are sources of toxins in the bodies of TTX-bearing organisms and in the surrounding environment in this site in the future.

On the other hand, since no reports about *A. floridus* and *H.* cf. *fasciata* were collected at the same time from the same site before, the relevance of the two TTX-bearing organisms needs to be verified. Many reports suggest that crustaceans such as crabs are the preferred prey of the octopuses, and Noguchi et al. hypothesized that the octopuses may prey on xanthid crabs [23,28,29,30,49,50]. From these reports, we also developed the hypothesis that there may be a predator–prey relationship between these two organisms at this site. Toxin resistance in *A. floridus* makes it difficult to paralyze it with TTX [51]. However, octopuses from the genus *Hapalochlaena* also possess other toxins such as maculotoxin, hapalotoxin, histamine, etc. [49] and use them both for offense [52] and defense [53], suggesting that they can effectively attack xanthid crabs.

The two TTX-bearing organisms used in this study had similar toxin compositions, although the exact toxification process remains unknown, suggesting that they belong to the same food web for TTX acquisition, including the same microbiomes, a similar diet, and/or a predator–prey relationship at this same site. Further studies are needed, including analyses of the stomach contents of both species and a search for toxic prey organisms living in these waters.

## 4. Materials and Methods

### 4.1. Sample Collection

Two types of TTX-bearing organisms, the xanthid crab *A. floridus* and the blue-lined octopus *H.* cf. *fasciata*, were collected from the same waters of Nagasaki, Japan. Specimens of *A. floridus* were collected in December 2020 and September and November 2021. Specimens of *H.* cf. *fasciata* were collected in September and November 2021. All specimens were transported to the laboratory of Nagasaki University and frozen at −20 °C until toxin extraction.

### 4.2. Toxin Extraction from A. floridus

TTX was extracted from the whole body of crab specimens in 2020 and the appendages in 2021, as previously described [26]. After homogenizing the sample, twice the volume of 0.1% acetic acid was added, and the mixture was heated in a boiling water bath for 10 min. The sample was cooled in ice water and then centrifuged at 3000× *g* for 10 min, and the supernatant was passed through an HLC-DISK membrane filter (0.45 µm, Kanto Chemical Co., Ltd., Tokyo, Japan). The obtained test solution was subjected to toxin quantification as described below.

### 4.3. Toxin Extraction from H. cf. fasciata

The blue-lined octopus *H.* cf. *fasciata* was divided into 6 parts as follows: buccal mass (including the anterior salivary glands, lips, and beak); PSGs; lateral mantle; viscera (including kidney, heart, and liver); gonads; and arms. Three volumes of 0.1% acetic acid were added for each 0.1 g of tissue; if less than 0.1 g of tissue, 300 μL was added. The test solutions were then prepared in the same manner as for the toxin extraction from the crabs.

### 4.4. Mouse Bioassay

The extracts of crabs in 2020 were examined using a mouse bioassay [54]. Lethal potency was determined by the time required to kill the mice and expressed in mouse units (MU). One MU is defined as the amount of toxin that kills a 20 g male ddY strain mouse within 30 min after intraperitoneal administration.

### 4.5. Purification of TTX and TTX Analogs

Extracts of *A. floridus* collected in 2020 were combined and purified according to the following procedure. The extract was defatted twice with an equal volume of dichloromethane, and the toxin in an aqueous layer was absorbed on activated charcoal, washed with water, and then eluted with 1% acetic acid/20% ethanol. The resulting eluate (2,160 MU) was submitted to a Bio-Gel P2 column and eluate with 0.03 mol/L acetic acid. The toxin fractions were submitted to a Bio-Rex 70 column, and toxins were separated with a linear gradient of 0–0.05 mol/L acetic acid. Finally, the toxic components were isolated using an ODS column (COSMOSIL(R) 5C18-AR-II Packed Column, 20 mm i.d × 250 mm, Nacalai Tesque, Inc., Kyoto, Japan).

### 4.6. Toxin Analysis by Instruments

#### 4.6.1. Liquid Chromatography-Tandem Mass Spectrometry for TTXs

LC-MS/MS was conducted according to a previously reported method [26]. LC was performed on an Alliance 2690 Separations Module (Waters Corp., Milford, MA, USA). A Mightysil RP-18 GP column (2.0 mm i.d × 250 mm, particle size 5 µm, Kanto Chemical Co., Inc.) was used with a mobile phase of 30 mM heptafluorobutyric acid in 1 mM ammonium acetate buffer (pH 5.0). The flow rate was set to 0.2 mL/min. The eluate was introduced into a Quattro micro API detector (Waters). TTX was ionized by positive-mode electrospray ionization with a desolvation temperature of 400 °C, a source block temperature of 150 °C, and a cone voltage of 50 V, and monitored at *m*/*z* 162 (quantitative) and *m*/*z* 302 (qualitative) as product ions (collision voltage 38 V) with *m*/*z* 320 as a precursor ion through a MassLynx NT operating system (Waters). TTX standard (purified from pufferfish ovary) was used as an external standard. The limit of detection (LOD) of TTX was 0.01 μg/g tissue (S/N = 3) and the limit of quantification (LOQ) of TTX was 0.03 μg/g tissue (S/N = 10). The other analogs were detected at *m*/*z* 336 > 162 for 11-oxoTTX, *m*/*z* 304 > 162 for deoxyTTXs, *m*/*z* 302 > 162 for 4,9-anhydroTTX, *m*/*z* 330 > 162 for 4-*epi*TTX and *m*/*z* 290 > 272 for 11-norTTX-6(R/S)-ol.

#### 4.6.2. High-Performance Liquid Chromatography with Fluorometric Detection for TTXs

HPLC-FLD for TTX analysis was conducted as described previously [55]. An Alliance 2690 Separation Module (Waters) connected with the Waters 2487 fluorescence detector. A Mightysil RP-18 column (4.6 mm i.d × 250 mm, particle size 5 µm, Kanto Chemical Co., Inc) was used. The mobile phase for the TTX and TTX analogs was 2 mM heptanesulfonic acid in 10 mM ammonium phosphate buffer (pH 7.0) at a flow rate of 1 mL/min. The eluate was continuously mixed with 4 M NaOH and heated at 110 °C. The intensity of the fluorescence was measured at 505 nm with 384 nm excitation. The same TTX standard was used to compare the retention time of TTX and TTX analogs. The LOD of TTXs was 0.02 μg/g tissue (S/N = 3) and the LOQ of TTXs was 0.06 μg/g tissue; (S/N = 10).

#### 4.6.3. High-Performance Liquid Chromatography with Fluorometric Detection for PSTs

PSTs analysis by HPLC-FLD was also conducted as previous methods [56]. An Alliance 2690 Separations Module (Waters) connected with a carbon Hypercarb^®^ column (4.6 mm i.d. × 100 mm length; 5 µm; Thermo Fisher Scientific, Waltham, MA, USA) was used. The column temperature was set at 25 °C and the separation of total 12 STX components was performed using two mobile phases—(A) 0.075% (*v*/*v*) TFA (Trifluoroacetic Acid) in water and (B) 0.025% (*v*/*v*) TFA in 50% (*v*/*v*) acetonitrile: water. The flow rate was set at 0.8 mL/min. Linear gradients were 96% A and 4% B to 75% A and 25% B over 30 min, then returned to 96% A and 4% B at 30.01 min and re-equilibrated for 8 min until the next injection. The eluate from the column was continuously mixed with 0.2 M KOH containing 1 M ammonium formate and 50% formamide with 50 mM periodic acid at a flow rate of 0.4 mL/min each and heated at 65 °C. The intensity of the fluorescence was measured at 392 nm with 336 nm excitation. PSTs mixed standard containing GTX1-4 and dcGTX2,3 were provided by the Japan Fisheries Research and Education Agency, and dcneoSTX, neoSTX, hyneoSTX, hySTX, dcSTX, and STX purified from the toxic crab *Zosimus aeneus* were used to identify. The LOD of STXs was 0.02–0.04 μg/g tissue (S/N = 3) and the LOQ was 0.06–0.12 μg/g tissue (S/N = 10).

## Figures and Tables

**Figure 1 toxins-15-00193-f001:**
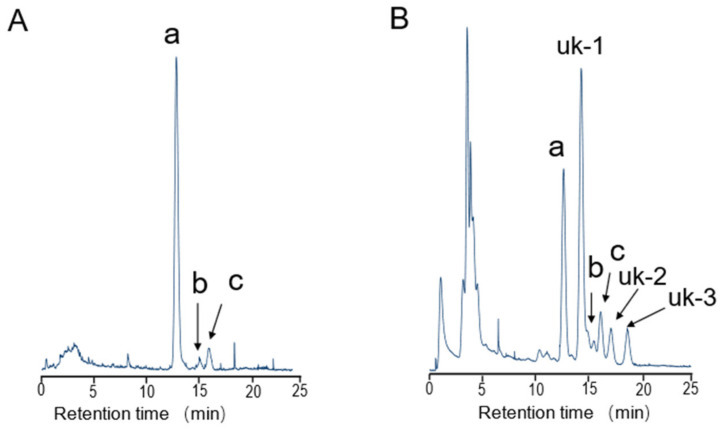
HPLC-FLD chromatograms of (**A**) TTX standard and (**B**) crab extracts in 2020. a: TTX; b: 4-*epi*TTX; c: 4,9-anhydroTTX.

**Figure 2 toxins-15-00193-f002:**
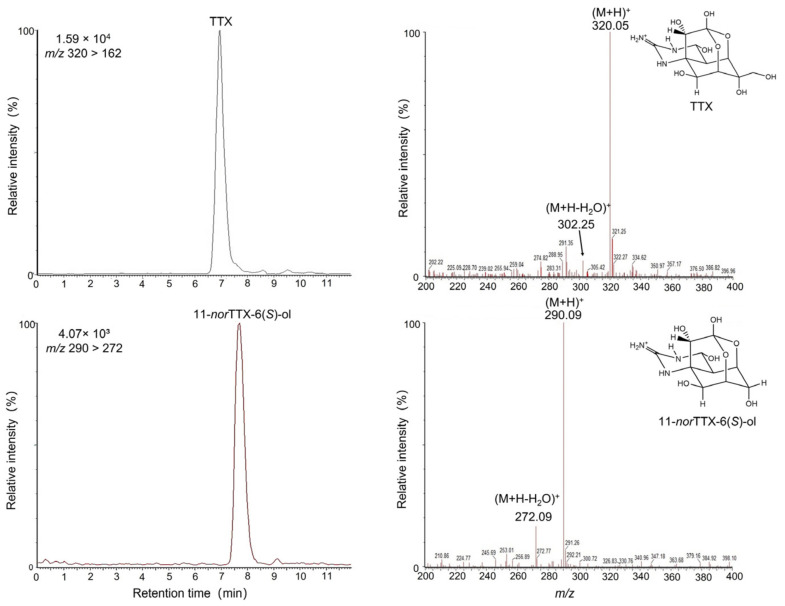
LC-MS/MS chromatograms and MS spectra of TTX (**upper**) and 11-*nor*TTX-6(*S*)-ol (**lower**) from crab extracts in 2020.

**Figure 3 toxins-15-00193-f003:**
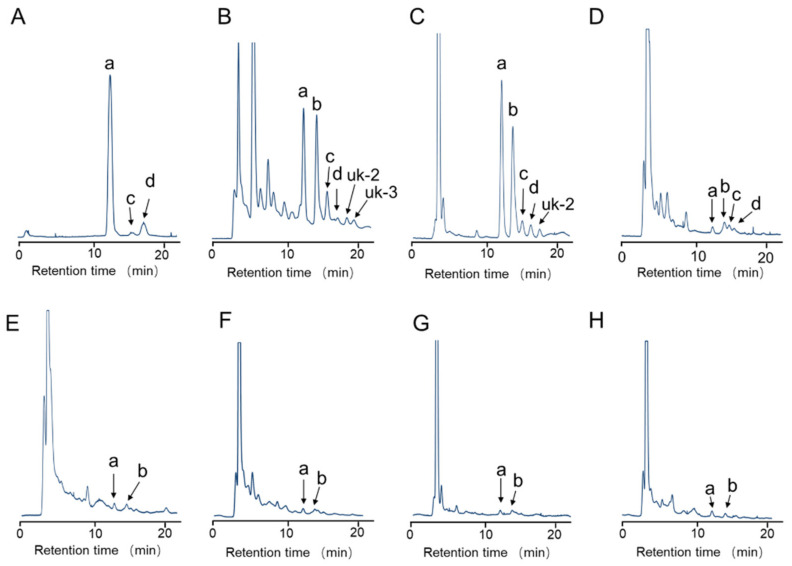
HPLC-FLD chromatograms of *A. floridus* extracts and each tissue of *H.* cf. *fasciata* (No. 3) collected in 2021. (**A**): TTX standard; (**B**): extracts of *A. floridus*; (**C**–**H**): tissues of *H.* cf. *fasciata*; C: PSGs; (**D**): buccal mass; (**E**): viscera; (**F**): gonads; (**G**): lateral mantle; (**H**): arms. a: TTX; b: 11-*nor*TTX-6(*S*)-ol; c: 4-*epi*TTX; d: 4,9-anhydroTTX.

**Figure 4 toxins-15-00193-f004:**
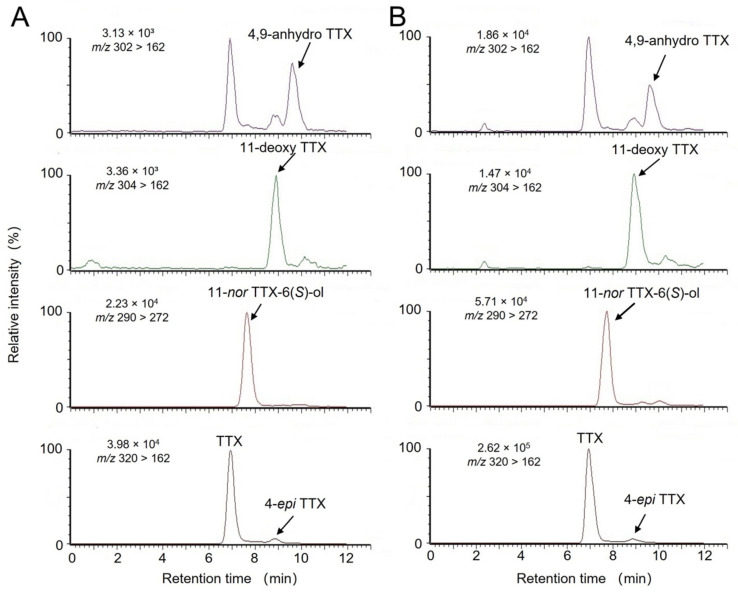
LC-MS/MS (MRM) chromatograms of (**A**) *A. floridus* extracts in 2021; (**B**) PSG of *H.* cf. *fasciata* collected in 2021.

**Table 1 toxins-15-00193-t001:** TTX concentrations in the whole body of *A. floridus* collected in December 2020.

Specimen No.	Body Weight (g)	TTX Concentration (μg/g)	Total TTX Amounts (μg)
1	4.42	10.73	47.42
2	3.77	1.05	3.95
3	2.96	0.73	2.16
4	11.41	11.88	135.53
5	11.59	11.13	128.94
6	5.16	0.60	3.10
7	9.69	15.22	147.44
8	9.51	10.52	100.05
9	9.28	45.46	421.88
Mean ± S.D	7.53 ± 3.4	11.93 ± 13.8	110.05 ± 131.3

**Table 2 toxins-15-00193-t002:** TTX concentrations in the appendages of *A floridus* collected in 2021.

Specimen No.	Collection Date	Body Weight (g)	TTX Concentration in Appendages (μg/g)
1		6.46	0.09
2		10.30	1.25
3		13.49	9.77
4		13.86	0.37
5		3.33	0.05
6	Sep. 2021	5.55	0.22
7		7.11	0.22
8		11.50	1.58
9		13.21	0.54
10		8.92	0.10
11		10.28	0.94
Mean ± S.D		9.45 ± 3.5	1.37 ± 2.8
1		17.15	0.15
2	Nov. 2021	10.08	25.54
3		13.64	0.39
4		11.86	0.09
Mean ± S.D		13.18 ± 3.7	6.54 ± 12.7

**Table 3 toxins-15-00193-t003:** TTX concentrations and total TTX amounts in each tissue of the blue-lined octopus *H.* cf. *fasciata*.

Number of Octopuses	Body Weight (g)	Collection Date	TTX Concentration (μg/g)						Average TTX Concentration (μg/g) ^1^	Total TTX Amounts (μg/Tissue)						Total (μg/Individual)
			PSGs	Viscera	Gonads	Buccal Mass	Lateral Mantle	Arms		Psgs	Viscera	Gonads	Buccal Mass	Lateral Mantle	Arms	
1	4.22	Sep. 2021	1.62	N.D ^2^	N.D	N.D	N.D	N.D	0.04	0.19	—	—	—	—	—	0.19
2	9.03	Nov. 2021	36.64	2.95	3.37	4.30	3.44	3.50	3.48	2.93	3.23	0.90	0.22	4.64	19.53	31.44
3	13.41		100.65	1.61	0.67	1.65	1.36	1.25	1.26	5.44	0.75	0.75	0.08	1.99	7.94	16.95
4	7.56		19.10	1.68	1.88	2.48	1.49	2.24	2.61	7.54	0.31	0.80	0.10	1.62	9.34	19.72
5	5.59		4.49	N.D	N.D	0.40	N.D	N.D	0.11	0.61	—	—	0.01	—	—	0.62
Mean	7.96		40.22	1.25	1.18	1.77	1.25	1.40	1.89	6.26	0.57	0.55	0.08	1.38	6.24	15.08
S.D	3.56		40.56	1.26	1.44	1.73	1.41	1.51	1.52	3.15	1.36	0.45	0.09	1.91	8.07	13.37

^1^ The average TTX concentration was calculated with total TTX amounts/body weight. ^2^ N.D.: ˂0.06 μg/g.

## Data Availability

The data that support the findings of this study are available from the corresponding author, upon reasonable request.

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
