# Peer review of "Tetrodotoxin Profiles in Xanthid Crab *Atergatis floridus* and Blue-Lined Octopus *Hapalochlaena* cf. *fasciata* from the Same Site in Nagasaki, Japan"

_toxins, 2023, doi:10.3390/toxins15030193_

Round 1
Reviewer 1 Report
Title: instead “same waters” set geographic region
Abstract has to be rewrite. Abstract has to include background and motivation to the paper, a brief description of the methods, the principal results, and then conclusions or interpretations”. For more information see https://www.mdpi.com/authors/layout#_bookmark5.
Line 20-22: TTX was found not only in “octopuses, newts, frogs and and crabs”. See last review in this area https://doi.org/10.3390/toxins14080576. Please rewrite this sentence.
Line 23-25: More 20 analogues of TTX were found in marine and terrestrial animals. Please rewrite this sentence.
Line 28 “the Pacific Ocean such as in Australia, Taiwan, the Philippines, and Japan”. Strange sentence – please rewrite.
Line 29: “blue-lined octopus of the genus Hapalochlaena”. Is there any other genus included in the concept of "blue-lined octopus". If not – please rewrite this sentence.
Line 100-103: Authors try to sum the total TTX amounts in the whole body using the TTX concentration in the appendages. But according Tsai et al (1997: https://doi.org/10.1016/S0041‐0101(97)00005‐6) in Atergatis floridus, the viscera and appendages were more toxic than other organs. Please rewrite this sentence.
Line 142: Authors use “toxin accumulations” term – but I think it is not correct. Authors investigated only TTXs concentration in different organs of marine animals. «toxin accumulations» proposed to proceed additional experiments with TTX-containing food or injection of toxins. To avoid misunderstanding I recommend replacing “toxin accumulations” to more correct term.
Line 142-147. This paragraph contains only Result information without any discussion with literature data. Authors should compare their results with same. For example see https://doi.org/10.3390/toxins15010016 or Ito, M et al (Toxins 2022, 14, 150) or Lopes et al (Toxicol. 2014, 146, 205–211)- where author proceed same investigations.
Line 158 – 159. Авторы путают понятие «exogenous» and «endogenous» TTX origin. TTX is bacterial origin (see doi:10.3390/toxins9050166), i.e. TTXs have an exogenous origin in TTX-bearing animals. As I understood, Authors used “exogenous” term as “food chain”, “endogenous” as from marine bacteria. Please correct «exogenous» and «endogenous» terms all over the MS.
Line 171-173. I disagree with this statement. For such a statement, Authors have to show a positive correlation between the toxicity of the salivary glands and the animal's weight. More over salivary glands is excretory organs which get nutrients (including toxins) from visceral area of octopuses (propose hemolymph) –i.e. by eating more TTX-containing objects, octopuses in general should become more toxic. More detailed information can be found in https://doi.org/10.3390/toxins14080576.
Line 176-177. The Authors proposed that TTX, localized in octopuse PSGs, is used for attack on a preys. But PSG are digestive glands the secret of which is not injected into the victim. PSGs are digestive glands for digesting food into disgestive tract. It is also not clear why the presence of TTXs in the hands used by octopuses for protection? Please explain both of these statements?
Line 178-196. This paragraph is very mixed , including many thoughts/ideas. Authors should completely revise this paragraph and try to focus on one idea: in the begining the authors talk about the analogues, then they talk about the presence of unknown analogues, and then they talk about the transformation of analogues. This paragraph should either be shortened or split into several.
Line 192-196: Sounds very ridiculous. I strongly recommend removing or completely rewriting sentences.
The first statement of the Authors that A. floridus is food for H. cf. fasciata, does not have any evidence. Therefore, to state that “the toxicity and toxin composition xanthid crab A. floridus and the blue-lined octopus H. cf. fasciata collected from the same site” are related to “a predator-prey relationship” – does not correct.
The second statement of the Authors (“the TTX-bearing prey organisms living in the Nagasaki waters are shared by A. floridus and H. cf. fasciata as the source of TTX”) – sounds more believable.
But the authors also forget that if TTX-producing bacteria are the primary source of TTX for TTX-bearing animals, then there is also a third variant of similar profiles - this is a common TTX-producing microbiome for similar regions. This idea was well wrought in https://doi.org/10.3390/toxins15010016. I strongly recommend that the authors remove the first statement and add the bacterial hypothesis.
Reviewer 2 Report
This paper is a relevant investigation into the concentrations and distribution of tetrodotoxin into two different marine organisms. Many TTX analogues are known, however, the absence of suitable certified reference materials makes it incredibly challenging if not impossible to identify the various analogues. For this reason, the profile information in the manuscript was unfortunately limited to only a small handful of TTX analogues, for which the authors cannot be faulted.
It would be beneficial, if possible, in future work to investigate and include additional TTX analogues which are suspected to be earlier precursors in the biosynthetic pathway including the trideoxy and dideoxy forms which may provide additional clarity on toxin profiles and improve understanding when comparing the similarities and differences.
Lines 21-22: It may be of value to acknowledge other species such as commonly consumed bivalves and gastropods which have also been reported internationally to be able to contain TTX, which has renewed discussions on regulation for food safety and the need for better understanding of the origin of TTX and how it distributes and accumulates along the food chain.
Line 35: Suggest change “3 toxic xanthid crabs [] are highly toxic” to “3 species of xanthid crabs [] are reported to be highly toxic”.
Line 42: May want to clarify that not all species of pufferfish are reported to be able to accumulate neurotoxins.
Lines 65-67: Was this analysis adjusted for recovery and/or suppression effects? LC-MS analysis of TTX can be influenced by substantial suppression from coeluting interferences which can be in high abundance even after carbon clean-up. Matrix effects can be variable from sample to sample which can result in underestimation.
Discussion: It would be worth mentioning some of the limitations around reference material availability that hinders the ability of ensuring peak identification and quantification of many of the TTX analogues, and further work that is needed in future TTX research.
Section 4.6. Not sure if the journal will allow the formatting, although I would recommend breaking the instrument methods into sub sections. 4.6.1. for LC-MS/MS, 4.6.2. for TTX HPLC-FL etc.
Round 2
Reviewer 1 Report
I fully agree with the authors comments. The authors advanced many hypotheses in the Discussion section. Usually, in such cases, one more section Conclusion and Future perspective is written . I would recommend to the authors to make such a section - but this is at their discretion.
My minor comment see below:
Line 210-212. In a recent work (doi:10.3390/md17120704), it was found that the spore culture of marine bacterium Bacillus sp. 1839 contains non-equilibrium analog the 5,6,11-trideoxyTTX. Please rewrite this sentence according to doi:10.3390/md17120704.
Author Response
Thank you for your valuable suggestion.
We have rewritten this sentence and added the reference you mentioned.
Please see P8 L 210-212 and P12 L 421-422.